# Serum Concentrations of Imidazole Dipeptides and Serum Amyloid A in a Bottlenose Dolphin (*Tursiops truncatus*) with Rhabdomyolysis: Potential Biomarkers for Muscular Damage

**DOI:** 10.3390/ani15131950

**Published:** 2025-07-02

**Authors:** Nanami Arakawa, Mika Otsuka, Takahisa Hamano, Momochika Kumagai, Sanae Kato, Takuya Hirai, Akira Yabuki, Osamu Yamato

**Affiliations:** 1Laboratory of Clinical Pathology, Joint Faculty of Veterinary Medicine, Kagoshima University, Kagoshima 890-0065, Japan; k9829543@kadai.jp (N.A.); yabu@vet.kagoshima-u.ac.jp (A.Y.); 2Kagoshima City Aquarium, Kagoshima 892-0814, Japan; m-otsuka@ioworld.jp (M.O.); m8023017@yahoo.co.jp (T.H.); 3Faculty of Fisheries, Kagoshima University, Kagoshima 890-0056, Japan; kumagai@fish.kagoshima-u.ac.jp (M.K.); kato@fish.kagoshima-u.ac.jp (S.K.); 4Departments of Veterinary Pathology, Faculty of Agriculture, University of Miyazaki, Miyazaki 889-2192, Japan; t-hirai@cc.miyazaki-u.ac.jp

**Keywords:** imidazole dipeptide, serum amyloid A, rhabdomyolysis, skeletal muscle, biomarker, bottlenose dolphin

## Abstract

Levels of imidazole dipeptides (IDPs) and serum amyloid A (SAA) were measured in the serum of a captive bottlenose dolphin affected by rhabdomyolysis. The IDP concentration was altered in response to skeletal muscle damage, as evaluated by the release of serum muscular enzymes, such as creatinine kinase and aspartate aminotransferase. The IDP concentration was altered similarly to the SAA concentration, when measured using an enzyme-linked immunosorbent assay specific to dolphin SAA and a latex turbidimetric immunoassay specific to human SAA. Accordingly, levels of IDP and SAA could be good biomarkers for detecting pathological damage or injury to skeletal muscles leading to myolysis and inflammation in dolphins.

## 1. Introduction

Rhabdomyolysis is a complex pathological condition involving the rapid dissolution of damaged or injured skeletal muscles [1,2,3]. In humans, rhabdomyolysis occurs due to trauma, compression, exertion, muscle hypoxia, changes in body temperature, drugs and toxins, infections, electrolyte imbalances, endocrine disorders, autoimmune disorders, and genetic defects [1,2,3]. The dissolution of skeletal muscles leads to the direct release of intramuscular components such as myoglobin, electrocytes, and enzymes, including creatinine kinase (CK), aspartate aminotransferase (AST), and lactate dehydrogenase, into systemic circulation [2,3].

The muscles of marine mammals, including cetaceans, have a high myoglobin content and net surface charge for oxygen storage and buffering capacity [4,5]. In cetaceans, a positive correlation has been observed between dive duration and myoglobin content in the skeletal muscles [6]. Accordingly, a high muscular myoglobin content is crucial in cetaceans, although it can also be a risk factor for rhabdomyolysis. This is because the large amount of released myoglobin is highly nephrotoxic and induces acute kidney injury (AKI), which can occur as a serious complication of rhabdomyolysis [7]. The released myoglobin causes proximal tubular cell death in the kidney, resulting in AKI. Rhabdomyolysis and secondary capture myopathy were observed in cetaceans that were stranded alive [8,9,10,11,12]. In a Japanese aquarium, a 5-year-old female bottlenose dolphin reportedly died due to rhabdomyolysis caused by trauma and subsequent AKI [13]. Therefore, an early diagnosis of rhabdomyolysis is crucial to ensure appropriate treatment and management of rhabdomyolysis in captive dolphins. Early diagnosis requires sensitive and specific biomarkers in addition to intramuscular enzymes such as CK and AST.

Imidazole dipeptides (IDPs) are a type of dipeptide comprising L-histidine containing an imidazole ring and β-alanine [14]. There are three major IDP analogs: anserine, carnosine, and balenine. IDPs are mostly found in the skeletal muscle of mammals, with balenine identified as the major IDP present in cetaceans. Thus, IDPs, especially balenine, could serve as a favorable marker of skeletal muscle damage and injury in dolphins.

Serum amyloid A (SAA) is an acute phase protein synthesized predominantly in the liver [15,16,17]. The concentration of SAA is known to increase after inflammatory reactions due to infection, tissue damage, trauma, and neoplasia in most tissues, including skeletal muscles. SAA is widely used as an inflammatory marker in various animal species and is reportedly valuable in dolphins [18]. Therefore, SAA may be useful in detecting the early phase of muscular damage or injury during the pathogenesis of rhabdomyolysis in dolphins.

In the current study, we measured IDPs and SAA using stored serum collected from a dolphin with rhabdomyolysis [13] and compared these two parameters with other common blood parameters to verify the relevance of these measurements in dolphins with muscle diseases.

Presently, the housing of marine mammals such as dolphins has become controversial among various stakeholders, leading to several problems, including legal issues, emotional and misleading messaging, impasses between different parties, and failures to consider scientific evidence [19,20]. Among these stakeholders, scientists and researchers should elucidate the factors that enhance the care and lives of marine mammals residing in facilities [20]. Therefore, the identification of novel biomarkers for marine veterinary diagnostics will contribute to the development of welfare science for captive dolphins.

## 2. Materials and Methods

### 2.1. Case Description

A 5-year-old female bottlenose dolphin (*Tursiops truncatus*) housed at the Kagoshima City Aquarium (Kagoshima, Japan) hit its body in a pool following a jump on 16 December 2018. The dolphin exhibited behavioral abnormalities, such as refusals to jump upon instructions and slow movements, on the same day. Serum myoglobin was detected on the same day and 2 days later. Subsequently, behavioral abnormalities gradually improved over several weeks.

Thereafter, anorexia, vomiting, and bradycardia were suddenly recorded on 21 February 2019 (Day 1, defined as the first day of illness), 67 days after the collision. On Day 1, the disease status rapidly worsened, with death occurring on Day 16 following markedly elevated concentrations of blood parameters, including IDPs and SAA. Following a postmortem autopsy and histological examination, the dolphin was diagnosed with rhabdomyolysis and secondary AKI. Changes in the clinicopathological data and the histopathological features have been described previously [13].

### 2.2. Blood Sample Collection

Blood samples were collected from the tail flukes using vacuum blood collection tubes. Heparinized plasma and serum were obtained by centrifugation and stored at −80 °C until use. This study utilized 27 blood samples and their clinicopathological data for the following time points: before the collision (Day −129); at the time of the collision and subsequent follow-up period (Days −67 to −15); and during the illness duration (Days 1 to 14).

### 2.3. Hematological and Biochemical Analyses

For hematological and biochemical analyses, blood samples were outsourced to a testing company (Clinical Pathology Laboratory, Inc., Kagoshima, Japan). Hematological data, including leukocyte (white blood cell [WBC]) counts, were obtained using cell counters (Sysmex XE-5000 and XE-2100; Sysmex Corporation, Kobe, Japan). Biochemical parameters, including CK and AST, were measured using automated biochemical analyzers (LABOSPECT 006 and 008; Hitachi, Tokyo, Japan), determined according to the Japan Society of Clinical Chemistry [21].

### 2.4. SAA Analysis

SAA concentrations were measured using two methods: an enzyme-linked immunosorbent assay (ELISA) specific to dolphin SAA [18] and a latex turbidimetric immunoassay (LTI) specific to human SAA [22,23], thereafter referred to as ELISA-SAA and LTI-SAA, respectively. SAA was measured using a dolphin-specific ELISA kit (SAA-18; Life Diagnostics, Inc., West Chester, PA, USA) according to the manufacturer’s instructions. This assay uses two peptide-specific dolphin SAA antibodies: one for solid-phase immobilization (microtiter wells) and another conjugated to horseradish peroxidase for detection. A typical standard curve provided an r value of 0.998. The serum was diluted by 2000- to 100,000-fold to avoid matrix effects before measurement. The absorbance of 3,3′,5,5′-tetramethylbenzidine was measured at 450 nm using a microplate reader (iMark; Bio-Rad, Richmond, CA, USA). For LTI-SAA measurement, we employed an automated biochemical analyzer (Pentra C200; HORIBA ABX SAS, Montpellier, France) with an SAA reagent specialized for human serum or plasma (LZ test ‘Eiken’ reagent; Eiken Chemical Co., Ltd., Tokyo, Japan). LTI-SAA concentrations were assessed using a calibrator (LZ SAA Calibrator Set; Eiken Chemical Co., Ltd., Tokyo, Japan), and a standard curve was constructed using the calibrator set.

### 2.5. IDPs Analysis

The concentrations of anserine, carnosine, and balenine were measured using a reversed-phase ion-pair high-performance liquid chromatography-ultraviolet method, as described previously [24]. The IDP concentration was expressed as the sum of the total concentrations of the three IDPs.

### 2.6. Statistical Analysis

For statistical analysis, Spearman’s correlation coefficients were used to determine the relationships between IDPs and other parameters (muscle markers [CK and AST] and SAA [LTI and ELISA]) using R version 4.5.0. In addition, the differenced series (difference between two time points) was also analyzed by Spearman’s correlation coefficients to evaluate the relationships of interlocking changes between IDPs and other parameters. *p* values of less than 0.05 were considered to indicate statistical significance. The appropriateness of the statistical analysis was evaluated by a professional statistical service (SATiSTA, Co., Ltd.; Uji, Japan).

## 3. Results

### 3.1. Changes in IDPs, CK, and AST Levels

The concentrations of anserine, carnosine, balenine, and total IDPs are presented in Appendix A. Among the total IDPs, the percentage of each IDP was as follows: anserine, 3.7–6.5%; carnosine, 6.9–25.4%; and balenine, 68.1–88.6%. Changes in the total IDP concentration and CK and AST activity are shown in Figure 1. On Day −129, when the dolphin was clinically healthy, the baseline IDP concentration, CK activity, and AST activity were 35.5 µM, 128, and 351 U/L, respectively. On Day −67, at the time of the collision, CK (623 U/L) and AST (470 U/L) activities were slightly elevated; however, the IDP concentration (31.7 µM) did not increase. Upon initiation of rhabdomyolysis, the IDP concentration (205.5 µM) first increased markedly on Day 3, whereas CK (197 U/L) and AST (407 U/L) activities were not notably elevated on the same day.

On Days 4 and 5, the IDP concentration temporarily decreased to twice the baseline level (69.5 µM), then steeply increased to a high level (295.4 µM) on Day 6, maintaining a high level until Day 9. In contrast, CK (1575 U/L) and AST (832 U/L) activities started to increase on Day 4, lagging behind the increase in IDP concentrations, and tended to increase until Days 7 (CK: 10,977 U/L) and 8 (AST: 3155 U/L). Subsequently, these activities gradually decreased until Day 9 (CK: 5664 U/L and AST: 2682 U/L).

On Day 10, the IDP concentration increased steeply to a higher level (563.0 μM); CK and AST activities increased synchronously with the increase in IDP levels. Furthermore, on Days 11 and 12, IDP, CK, and AST showed a tendency toward elevation, reaching the highest level on Days 13 (AST: 3559 U/L) and 14 (IDP: 629.7 µM and CK: 26,231 U/L), when the dolphin developed a terminal condition.

### 3.2. Changes in SAA and WBC Levels

The changes in ELISA- and LTI-SAA concentrations and WBC counts are shown in Figure 2. On Day −129, the baseline values of ELISA-SAA, LTI-SAA, and WBC count were 12.3 mg/L, 0.00 mg/L, and 5700 /µL, respectively. On Day −67, at the time of the collision, the WBC count (12,300/µL) showed a temporal increase, accompanied by a slight increase in ELISA-SAA (74.0 mg/L) and LTI-SAA (0.24 mg/L) concentrations. Following the initiation of rhabdomyolysis, these three parameters started to increase on Day 3 (ELISA-SAA: 327.3 mg/L, LTI-SAA: 5.25 mg/L, and WBC: 6700/µL).

Both SAA concentrations decreased temporarily on Days 4 and 5, followed by a steep increase on Day 6 (ELISA-SAA: 368.1 mg/L and LTI-SAA: 16.86 mg/L). The elevated ELISA-SAA level was retained until Day 9, further increasing to a higher level on Days 10 (810.7 mg/L) and 12 (843.1 mg/L). On Day 6, the LTI-SAA increased persistently, peaking on Day 14 (84.81 mg/L). The WBC count followed a similar pattern to both SAAs, with the highest count observed on Day 14 (2000/µL).

Compared with all parameters in Figure 1 and Figure 2, IDP and SAA concentrations exhibited markedly similar up-and-down patterns, as they showed four steep increases on almost the same days: Days 3, 6, 10, and 11 or 12.

### 3.3. Relationship Between IDPs and Other Parameters

The relationships (correlation coefficient and *p* value) between the IDP concentration and other parameters (A: CK, B: AST, C: ELISA-SAA, and D: LTI-SAA) are shown in Figure 3. Strong positive (r = 0.853–0.930) and significant (*p* < 0.001) correlations were detected between the IDP concentration and all other parameters.

The relationships (correlation coefficient and *p* value) of the differenced series between the IDP concentration and other parameters (A: CK, B: AST, C: ELISA-SAA, and D: LTI-SAA) are shown in Appendix A. A positive (r = 0.690) and significant (*p* < 0.001) correlation was detected between the IDP concentration and LTI-SAA (D). However, there were no significant correlations between IDP and the other parameters (A: CK, B: AST, and C: ELISA-SAA).

## 4. Discussion

Early diagnosis and management of rhabdomyolysis can prevent potential complications such as AKI and reduce mortality [3]. Thus, biomarkers for detecting skeletal muscle damage or injury are crucial for assessing the disease’s onset, real-time progression, severity, and prognosis. These biomarkers include CK and AST, which are common serum biochemical parameters. Serum CK level is one of the most important diagnostic and monitoring parameters for rhabdomyolysis and the most sensitive and specific indicator of skeletal muscle damage or injury among clinicobiochemical parameters [2]. In humans, the serum CK level rapidly changes according to the level of muscle damage or injury, with levels declining by 40–50% each consecutive day if the damage is not developed, given the relatively short half-life of CK (approximately 48 h). In contrast, AST is widely distributed in the body, mainly in the liver and muscles, and exhibits slower response kinetics than CK [3]. In human cases of rhabdomyolysis, AST peaks at 3 to 4 days, returning to baseline levels between 6 and 10 days if the condition is resolved [25]. Despite the lack of specificity and sensitivity of AST, the accuracy of the diagnosis and assessment of rhabdomyolysis can be improved by combining AST with CK [3]. Nevertheless, additional ideal biomarkers are needed to ensure more accurate management of muscle damage or injury.

In this study, we found that IDP concentrations increased in the dolphin with rhabdomyolysis in a manner similar to common muscle markers, including CK and AST activities (Figure 1). Following rhabdomyolysis initiation, the IDP concentration started increasing one day before the increase in CK and AST activities. We detected a significant positive correlation, with high coefficients, between IDP and the activity of these two muscle enzymes (Figure 3). These findings suggest that IDP could be a sensitive biomarker for detecting skeletal muscle damage and injury. However, the IDP concentration was unaltered when the dolphin collided with the poolside on Day −67, although CK and AST activities increased slightly. This may indicate that IDP may be released from seriously damaged or injured skeletal muscles, such as those with severe inflammation and necrosis. In addition, IDP may have a shorter half-life than CK and AST. This could be explained by the steep increase in the IDP concentration on Days 3, 6, 10, and 11 when muscle damage and inflammation may have developed further, and with the concentration declining or remaining constant, following the steep increases, in the absence of muscle damage and inflammation. The half-life difference between IDP and the two muscle enzymes may reflect a timing discrepancy in the changes in IDP concentration and enzyme activities, resulting in no significant correlation in the differenced series between them (Appendix A).

During the collision on Day −67, the WBC count increased temporarily, and SAA concentration, especially ELISA-SAA, increased slightly in response to the blow on a restricted muscle area, potentially resulting in mild to moderate injury and subsequent temporal inflammation. However, whether the collision-induced trauma led to the initiation of rhabdomyolysis two months later could not be established. Intramuscular scars that form after the first muscle injury and inflammation may self-destruct with intense exercise and/or muscle hypoxia. This type of recurrent injury may cause active inflammation, resulting in the initiation and progression of rhabdomyolysis. Although rhabdomyolysis is caused by diverse factors, the core mechanism is intracellular Ca^2+^ overload [3]. Intracellular Ca^2+^ overload induces the reduction in ATP and production of reactive oxygen species via the deterioration of complex pathways, which further exacerbate damage and extend the area of rhabdomyolysis [1,26]. If the collision on Day −67 serves as a trigger for subsequent rhabdomyolysis, it may be necessary to treat the collision-induced muscular damage and prevent subsequent deterioration for a sufficiently prolonged period. For example, anserine as a muscle protectant, vitamins E and C as antioxidants, and anti-inflammatory agents may be effective for preventing subsequent deterioration and the onset of systemic rhabdomyolysis, based on the core mechanism of this disease.

Recently, we encountered another 15-year-old female bottlenose dolphin at a different aquarium that collided with the pool after jumping. CK (360 U/L) and AST (311 U/L) activities increased slightly on the day after the collision compared with the levels observed one month post-collision (CK, 123 U/L; AST, 199 U/L). However, IDP concentration (32.9 µM) was not elevated compared with the level observed one month post-collision (28.6 µM) and the average level (31.8 µM) obtained from 10 clinically healthy bottlenose dolphins [24]. These data are markedly similar to those obtained with subsequent rhabdomyolysis observed in this study. The day after the collision, the dolphin was orally administered anserine (90 mg twice daily, Marine Active 10, Yaizu Suisankagaku Industry Co., Ltd., Shizuoka, Japan), as well as adequate vitamins E (700 mg once daily) and C (4000 mg once daily) supplementation for two months post-collision. As a result, the dolphin did not develop subsequent deterioration or rhabdomyolysis, suggesting that these muscle protectants and antioxidative agents were effective in treating temporal muscle damage and injury and preventing subsequent systemic rhabdomyolysis. Further research is required to identify therapeutic and preventive strategies for dolphins and other animals with muscle damage or injury.

Rhabdomyolysis in captive dolphins residing in aquariums and institutes has rarely been reported; therefore, ours may be the first reported captive case that was analyzed in detail clinically, clinicopathologically, and histopathologically [13]; however, we have personally heard of other cases of suspected rhabdomyolysis in captive dolphins. In addition to our case, almost all reported cetacean cases with rhabdomyolysis or capture myopathy were live-stranded cetaceans, including various dolphins and whales that were not cured and eventually died [8,9,10,11,12]. The histopathological findings in all these cases [8,9,10,11,12] and the clinicopathological findings obtained from the blood samples of three cases [10,11,12] indicated that all the stranded cetaceans had severe skeletal muscle damage and complications, including myoglobinuric nephrosis, AKI, and cardiac dysfunction. Based on these findings, it would have been difficult to cure these animals. However, early diagnosis and treatments using muscle protectants and antioxidative agents may cure mild or moderate cases and contribute to increasing the survival rates of live-stranded cetaceans, possibly improving the welfare of captive dolphins and wild cetaceans. Further studies are required to evaluate the efficacy of these treatments in live-stranded cases.

In humans, carnosinase is present at high concentrations in the blood [14]. Carnosine absorbed from the intestinal tract is broken down by carnosinase into L-histidine and β-alanine, and carnosine is re-synthesized in skeletal muscles. Consequently, blood levels of carnosine are usually low despite carnosine being the main IDP in human muscles. However, carnosinase is thought to be absent in nonhuman primates. Blood carnosine levels are reportedly elevated during external rhabdomyolysis syndrome in horses [27]. The blood levels of carnosine are lower in foals than in adult horses because foals have underdeveloped muscles. Therefore, the metabolism and function of IDPs, particularly in animals, are yet to be elucidated. Further studies are needed to clarify how IDPs leak from the skeletal muscles of dolphins into the blood and how the released IDPs are degraded or absorbed into the muscles.

In the current study, we measured SAA concentrations, along with a common inflammatory marker, the WBC count, in a dolphin with rhabdomyolysis (Figure 2). In particular, both ELISA- and LTI-SAA concentrations showed an up-and-down change at the time of collision on Day −67 and after rhabdomyolysis initiation on Days 3 to 14, in accordance with the WBC count, suggesting that SAA is a good and sensitive biomarker for acute and active inflammation. In addition, both ELISA- and LTI-SAA concentrations were changed similarly to the IDP concentration and WBC count (Figure 1 and Figure 2). A significant positive correlation with high coefficients was detected between the IDP and two SAA concentrations (Figure 3). In addition, a significant positive correlation was also detected in the differenced series between the IDP and LTI-SAA (Appendix A). Both SAA concentrations changed in synchronization with the changes in the IDP concentration. This suggests that IDPs are released from muscles accompanied by active inflammation, and that the half-lives of IDPs and SAA may be similar. SAA is one of the major acute phase proteins, along with C-reactive protein and haptoglobin [28]. Typically, the concentrations of these proteins are very low under healthy conditions and dramatically increase up to 1000-fold following inflammatory stimuli. The SAA level peaks at approximately 24–48 h and then decreases rapidly upon recovery and improvement. Our data suggests that the IDP concentration may be altered hourly, similar to SAA.

The role of SAA as an early marker of inflammation in marine mammals, including dolphins, has been demonstrated previously [29]. However, it is difficult to accurately determine these concentrations owing to a lack or poor availability of species-specific reagents. In this study, we compared ELISA-SAA with LTI-SAA, revealing that ELISA-SAA is more sensitive to the suspected inflammatory status than LTI-SAA, as shown in Figure 2. This is because the ELISA employed was specific for dolphin SAA. The sensitive up-and-down changes in ELISA-SAAs rather than in LTI-SAAs might have resulted in no significant correlation between the differenced series of IDP and ELISA-SAAs (Appendix A). Although LTI-SAA was originally used to measure human SAA, an antibody in this LTI reagent can cross-react incompletely with dolphin SAA [22]. This LTI reagent has been used to measure SAA in horses [24,30], cats [30,31], dogs [30], deer [32], and dolphins [29]. Recently, another LTI reagent for domestic animals (VET-SAA) was developed by the same company that produced the human LTI reagent used in the present study. VET-SAA is available for the measurement of SAA in most domestic animals, including dogs [33], cats [31], cattle [34,35], horses [23], pigs [36], and rabbits [37]; however, the antibody in this reagent does not cross-react with dolphin SAA. LTI-SAA was not as sensitive as ELISA-SAA in this study; however, the LTI reagent can be installed in various automated biochemical analyzers, and the SAA concentration can be measured rapidly and easily. Therefore, LTI-SAAs are more practical and available at clinical sites. ELISA- and LTI-SAA should be flexibly applied based on the situation and individual advantages.

## 5. Conclusions

Collectively, the findings of this study revealed that the serum concentrations of IDPs and SAA increased with disease progression in a bottlenose dolphin with rhabdomyolysis. These data suggest that IDP concentration could be employed to detect muscle damage and injury, such as necrosis and inflammation, in dolphins. Thus, IDPs and SAA can be utilized in dolphins in practical situations. However, the data in this study being from a single case prevented this observation from being fully generalizable; therefore, further studies are required to confirm the clinical utility of IDPs and SAA by increasing the sample size of cases with muscular diseases.

## Figures and Tables

**Figure 1 animals-15-01950-f001:**
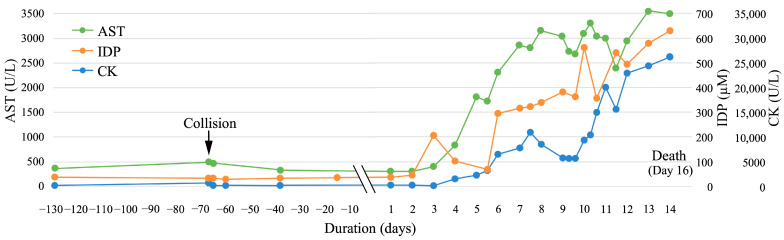
Changes in total imidazole dipeptides (IDPs: orange line) concentration, creatine kinase (CK: blue line) activity, and aspartate aminotransferase (AST: green line) activity in a 5-year-old female bottlenose dolphin with rhabdomyolysis that finally died on Day 16. The arrow indicates the time of collision (Day −67).

**Figure 2 animals-15-01950-f002:**
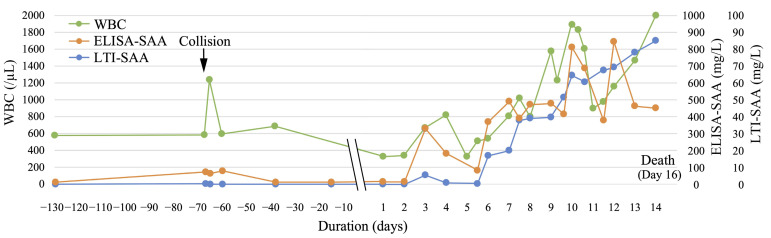
Changes in serum amyloid A (SAA) concentration measured using an enzyme-linked immunosorbent assay specific to dolphin SAA (ELISA-SAA: orange line), SAA concentration measured by a latex turbidimetric immunoassay specific to human SAA (LTI-SAA: blue line), and leukocyte (WBC: green line) count in a 5-year-old female bottlenose dolphin with rhabdomyolysis that finally died on Day 16. The arrow indicates the time of collision (Day −67).

**Figure 3 animals-15-01950-f003:**
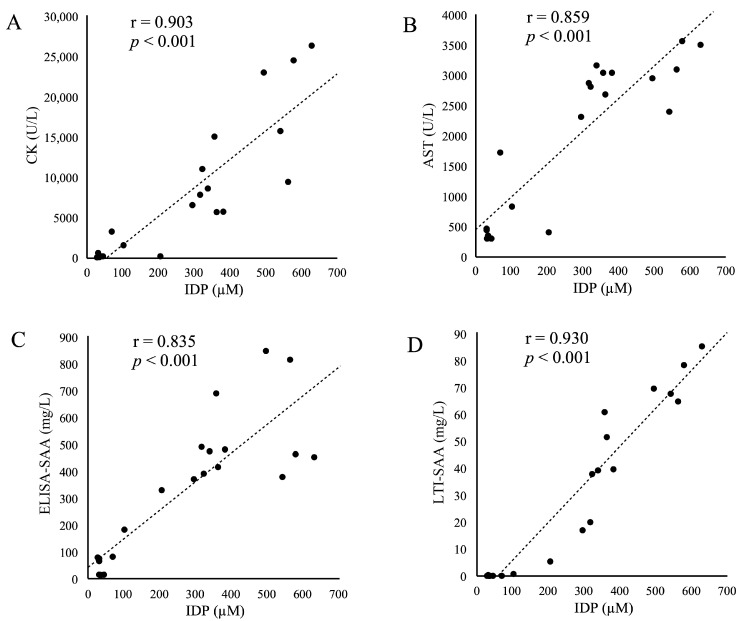
Correlation coefficient (r) and *p* value between total imidazole dipeptides (IDPs) concentration and other parameters: (**A**) creatine kinase (CK) activity; (**B**) aspartate aminotransferase (AST) activity; (**C**) serum amyloid A (SAA) concentration measured using an enzyme-linked immunosorbent assay specific to dolphin SAA (ELISA-SAA); and (**D**) SAA concentration measured using a latex turbidimetric immunoassay specific to human SAA (LTI-SAA). The dots represent the observations and broken lines represent the linear regression models for significant correlations.

## Data Availability

The original contributions presented in this study are included in the article/Appendix A. Further inquiries can be directed at the corresponding author.

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
