# Peer review of "Serum Concentrations of Imidazole Dipeptides and Serum Amyloid A in a Bottlenose Dolphin (Tursiops truncatus) with Rhabdomyolysis: Potential Biomarkers for Muscular Damage"

_animals, 2025, doi:10.3390/ani15131950_

Round 1

Reviewer 1 Report

Comments and Suggestions for Authors

Title: I recommend reshuffling some words in the title to make it more catchy.

Lime 47-52: I assume these are quoted based on literature; authors must credit the literature here.

Introduction: Authors must add a brief to the studied species, why this study is important for their welfare.

Objectives and aims of the research are not clear and well-defined anywhere in the intro.

Section 2.1: Is it clear authors used a single animal for the whole study?  A table with a details record of clinical pathology must be added as a supplement. In addition, I just wonder why authors didn’t compare this value with a healthy animal, to make the results more rigorous and statistically sound.

Section 2.4:  ELISA require adding more information regarding the processing of the blood, antibodies, and secondary antibodies, substratum/chromogen used. Also author may wish to mention the standard curve R value for a proper idea.

Section 2.5: as mentioned before, this protocol must be added with more information, including sample processing and the details procedure of the quantification.

Results: Still, it’s not clear anywhere how many samples ( even false replicates) were used for analysis.

Conclusion: This study has quite remarkable limitations; authors must include a brief on the limitations of current research and what else can be done to ensure more accuracy and legitimate results in such research arena.

Well, as far as I understand, authors used a single animal for the whole study, which suited for case report very well. I am recommending a major revision and suggestions for consideration as a case report.

Author Response

Responses to the comments from Reviewer 1 (We also attached the word file that is the same as the following description)

Dear Reviewer 1,

First of all, we appreciate the time the editor and reviewers have taken to read and review our manuscript. Their valuable comments have significantly improved several aspects of our paper. The following document presents our responses to comments and suggestions from the reviewers. It includes the original comments in italics and blue and the subsequent responses we made. The revised parts are indicated in red.

Comments and Suggestions for Authors

Title: I recommend reshuffling some words in the title to make it more catchy.

Authors’ response: Thank you for your advice. If we had come up with a catchy title, it might have given the impression of being shallow. This case study is based on a single case, which is one of the limitations in this study. We did not want to give the readers the impression that we overstated the findings by the title. In addition, since we are not native English speakers, it is very difficult to give it a kind of "catchy" title, which may give the readers the impression of being shallow. That is because we chose the current simple title that describes facts and the possibility of utility. We appreciate your understanding.

Lime 47-52: I assume these are quoted based on literature; authors must credit the literature here.

Authors’ response: Thank you for the reviewer's suggestion. We intended to credit the first sentence by the three review papers (ref. 1–3), and further credit the following two sentences by the same references without filling out; however, that actually caused the misunderstanding. We added the reference numbers after the second and third sentences according to the reviewer's request in the first paragraph of Introduction section.

Introduction: Authors must add a brief to the studied species, why this study is important for their welfare.

Authors’ response: Thank you for the reviewer's valuable suggestion. We are happy to add the description about animal welfare, one of the ultimate goals, in our researches about captive dolphins. We added a paragraph below as the last paragraph of the Introduction section.

"Presently, the housing of marine mammals such as dolphins has become contro-versial among various stakeholders, leading to several problems, including legal issues, emotional and misleading messaging, impasses between different parties, and the failures to consider scientific evidence [19, 20]. Among these stakeholders, scientists and researchers should elucidate the factors that enhance the care and lives of marine mammals residing in facilities [20]. Therefore, the identification of novel biomarkers for marine veterinary diagnostics will contribute to the development of welfare science for captive dolphins."

In addition, in response to this additional ultimate goal (welfare), we added an additional discussion about the welfare in the Discussion section. Another reviewer (Reviewer 2) requested us to add the discussion more about other cetaceans with rhabdomyolysis; therefore, we added another paragraph in the Discussion section in which we described the welfare as well. Please see the Discussion section.

Objectives and aims of the research are not clear and well-defined anywhere in the intro.

Authors’ response: Thank you for the reviewer's comment. The original minimal objective is to verify the relevance of these measurements (IDPs and SAA) in dolphins with muscle diseases (rhabdomyolysis). This was already described in the original last paragraph of the Introduction section. In the revised manuscript, we added an additional paragraph about the welfare as the last paragraph of the Intro according to the reviewer's request, which seems to be one of the ultimate goals or objectives in our study. We believe that the objectives and aims of our study have been clearer by the additional description about the animal welfare than that in the original manuscript. Thank you again for the reviewer's suggestion about the welfare science.

Section 2.1: Is it clear authors used a single animal for the whole study?  A table with a details record of clinical pathology must be added as a supplement. In addition, I just wonder why authors didn’t compare this value with a healthy animal, to make the results more rigorous and statistically sound.

Authors’ response: Thank you for the reviewer’s concerns that are very valuable for us. We do not think that data from a single animal is enough. There is the limitation and we got a similar comment and suggestion from another reviewer (Reviewer 2). Therefore, we additionally described the limitation, exploratory nature of the findings, and need of follow-up studies in the Conclusion section as follows.

"However, the data from a single case in this study prevented this observation from being fully generalizable; therefore, further studies are required to confirm the clinical utility of IDPs and SAA by increasing the sample size of cases with muscular diseases."

As described in our manuscript, we already reported this case as "case report" showing histopathological findings and clinicopathogical data in the paper (ref. 13, Table 1) published in 2022. Furthermore, in our methodological paper about IDPs measurement (ref. 24, Table 5) published in 2021, we indicated the serum concentrations (means ± SD) of anserine, carnosine, and balenine in clinically healthy bottlenose dolphins (n = 10). The present case study was performed as a retrospective study following our case report (ref. 13) and methodological study (ref. 24). These three papers are a series of our study regarding IDPs and SAA in dolphins with muscular damage. In the future, we will collect the data from dolphins with muscular diseases and other diseases to contribute to marine veterinary medicine and welfare in dolphins and cetaceans. We added this averaged level of IDPs obtained from 10 heathy dolphins in our previous research (ref. 24) because this additional description makes the reader know that the reliable reference value in healthy dolphins is already reported for the comparison. We appreciate your understanding.

"However, IDP concentration (32.9 µM) was not elevated compared with the level observed one month post-collision (28.6 µM) and the averaged level (31.8 µM) obtained from 10 clinically healthy bottlenose dolphins [24]."

Section 2.4: ELISA require adding more information regarding the processing of the blood, antibodies, and secondary antibodies, substratum/chromogen used. Also author may wish to mention the standard curve R value for a proper idea.

Authors’ response: Thank you for the reviewer’s comment about ELISA-SAA measurement. We added the description according to the reviewer's request as follows. In addition, we added a name of chromogen "3,3′,5,5′-tetramethylbenzidine" as well.

"This assay uses two peptide-specific dolphin SAA antibodies: one for solid-phase im-mobilization (microtiter wells), and another conjugated to horseradish peroxidase for detection. A typical standard curve provided an R value of 0.998."

Section 2.5: as mentioned before, this protocol must be added with more information, including sample processing and the details procedure of the quantification.

Authors’ response: Thank you for the reviewer's comment. This measurement of IDPs was newly developed by us in 2021 (reference no. 24 in the revised manuscript) as our proceeding study. The method includes a moderately complicated procedure having a number of steps. Therefore, we cannot describe this method briefly in this manuscript. The readers can reproduce the measurement according to the total method and equipment described in that paper (ref. 24). We appreciate your understanding.

Results: Still, it’s not clear anywhere how many samples (even false replicates) were used for analysis.

Authors’ response: Thank you for your comment. We had already described the exact number (27) of blood samples in "2.2. Blood sample collection" of the Materials and Methods section. Each parameter lacks one or two data because of a shortage of the sample volume for measurement, but we exactly illustrated dots (number and position) in the figures (Figures 1 and 2). We appreciate your understanding.

Conclusion: This study has quite remarkable limitations; authors must include a brief on the limitations of current research and what else can be done to ensure more accuracy and legitimate results in such research arena.

Authors’ response: We appreciate the valuable comment. As mentioned above, we additionally described the limitation, exploratory nature of the findings, and need of follow-up studies in the Conclusion section as follows.

"However, the data from a single case in this study prevent this observation from being fully generalizable; therefore, further studies are necessary to confirm the clinical utility of IDPs and SAA by accumulating the number of cases with muscular diseases."

Well, as far as I understand, authors used a single animal for the whole study, which suited for case report very well. I am recommending a major revision and suggestions for consideration as a case report.

Authors’ response: We appreciate the reviewer's comment. As we mentioned before, the present study was performed as a retrospective study not a case report following our two related papers (ref. 13 and 24) in order to find new biomarkers (IDPs and SAA) for muscular damage. We appreciate your understanding.

Reviewer 2 Report

Comments and Suggestions for Authors

Peer Review Report of the article titled "Serum Concentrations of Imidazole Dipeptides and Serum Amyloid A in a Bottlenose Dolphin (Tursiops truncatus) with Rhabdomyolysis: Potential Biomarkers for Muscular Damage":

  1. General Assessment

This manuscript presents a compelling case study that evaluates serum imidazole dipeptides (IDPs) and serum amyloid A (SAA) concentrations as potential biomarkers of rhabdomyolysis in a bottlenose dolphin. The topic is relevant and important for veterinary diagnostics, especially within marine mammal medicine. The manuscript is generally well-written, methodologically sound, and offers original data that could have clinical utility.

  1. Strengths
  • Novelty: The study addresses a novel and underexplored area — the use of IDPs as early biomarkers of muscle damage in marine mammals.
  • Methodological clarity: The use of validated biochemical assays (ELISA, LTI, HPLC) enhances the robustness of the measurements.
  • Correlative analysis: The statistical comparison between IDPs, CK, AST, and both forms of SAA offers strong support for IDPs as a viable biomarker.
  • Figures and Data: The time-series data clearly illustrate the temporal relationships between biomarker elevations and disease progression.

  1. Weaknesses and Suggestions for Improvement

3.1. Scope and Interpretation

  • Case Study Limitation: The findings are based on a single case. While the case is detailed, this inherently limits generalizability. The authors do mention another dolphin case with comparative data, but a stronger comparative or control group analysis would increase reliability.
    • Suggestion: Emphasize the exploratory nature of the findings and propose follow-up studies involving larger sample sizes.

3.2. Statistical Analysis

  • Correlation of Differenced Series: The differenced series analysis shows weaker correlations than the absolute values. The implications of this should be discussed more deeply.
    • Suggestion: Briefly explain the biological significance or possible reasons for the disparity between absolute and differenced series correlations.

3.3. Literature Integration

  • Comparative Context: While the discussion does reference analogous human and equine conditions, it would benefit from more integration with broader marine mammal literature.
    • Suggestion: Include some reference ranges for the different markers (in all type of animals) and add more comparative data from similar studies in other cetaceans or pinnipeds, if available.

  1. Ethical and Transparency Standards
  • Ethical approval is appropriately noted.
  • Funding and potential conflicts of interest are transparently disclosed.

  1. Conclusion and Recommendation

The manuscript presents important preliminary findings on the potential of IDPs and SAA as early biomarkers for skeletal muscle damage in dolphins. While limited by its single-subject design, the study is a valuable contribution to marine veterinary diagnostics.

P.S.

Please make sure that the single error that I found in the article (pdf attached) is addressed

Author Response

Responses to the comments from Reviewer 2 (We attached the word file that includes tye same description as follows)

Dear Reviewer 2,

First of all, we appreciate the time the editor and reviewers have taken to read and review our manuscript. Their valuable comments have significantly improved several aspects of our paper. The following document presents our responses to comments and suggestions from the reviewers. It includes the original comments in italics and blue and the subsequent responses we made. The revised parts are indicated in red.

Comments and Suggestions for Authors

Peer Review Report of the article titled "Serum Concentrations of Imidazole Dipeptides and Serum Amyloid A in a Bottlenose Dolphin (Tursiops truncatus) with Rhabdomyolysis: Potential Biomarkers for Muscular Damage":

General Assessment

This manuscript presents a compelling case study that evaluates serum imidazole dipeptides (IDPs) and serum amyloid A (SAA) concentrations as potential biomarkers of rhabdomyolysis in a bottlenose dolphin. The topic is relevant and important for veterinary diagnostics, especially within marine mammal medicine. The manuscript is generally well-written, methodologically sound, and offers original data that could have clinical utility.

Authors’ response: Thank you very much for highly evaluating our paper. We are very grateful to have your valuable comments and suggestions. We carefully revised our paper according to your professional comments.

Strengths

Novelty: The study addresses a novel and underexplored area — the use of IDPs as early biomarkers of muscle damage in marine mammals.

Methodological clarity: The use of validated biochemical assays (ELISA, LTI, HPLC) enhances the robustness of the measurements.

Correlative analysis: The statistical comparison between IDPs, CK, AST, and both forms of SAA offers strong support for IDPs as a viable biomarker.

Figures and Data: The time-series data clearly illustrate the temporal relationships between biomarker elevations and disease progression.

Authors’ response: We appreciate the reviewer's assessment on our study's strengths. We are so happy that the reviewer found all of our aims in this study. I truly feel that all my efforts on this case were worth it. However, regardless of that, we carefully revised our paper according to the reviewer's comments and suggestions on our study's weak points as follows.

Weaknesses and Suggestions for Improvement

3.1. Scope and Interpretation

Case Study Limitation: The findings are based on a single case. While the case is detailed, this inherently limits generalizability. The authors do mention another dolphin case with comparative data, but a stronger comparative or control group analysis would increase reliability.

Suggestion: Emphasize the exploratory nature of the findings and propose follow-up studies involving larger sample sizes.

Authors’ response: Thank you for your concerns that are valuable for us, and we additionally described the limitation, exploratory nature of the findings, and need of follow-up studies in the conclusion section as follows.

"However, the data from a single case in this study prevented this observation from being fully generalizable; therefore, further studies are required to confirm the clinical utility of IDPs and SAA by increasing the sample size of cases with muscular diseases."

3.2. Statistical Analysis

Correlation of Differenced Series: The differenced series analysis shows weaker correlations than the absolute values. The implications of this should be discussed more deeply.

Suggestion: Briefly explain the biological significance or possible reasons for the disparity between absolute and differenced series correlations.

Authors’ response: Thank you for the reviewer’s comment about this issue. We feel sorry that I omitted this explanation in this paper because we explained the difference of half-life in each parameter instead, which may cause no significant correlation between the IDP and two muscular enzymes. Furthermore, we assumed that the more sensitive up-and-down changes in ELISA-SAA than in LTI-SAA resulted in no significant correlation between the IDP and ELISA-SAA, although both SAA concentrations changed in synchronization with the changes in the IDP concentration.

We added these explanations at the end of the second paragraph and in the middle of the last paragraph in the Discussion section as follows.

"The half-life difference between IDP and the two muscle enzymes may reflect a timing discrepancy in the changes in IDP concentration and enzyme activities, resulting in no significant correlation in the differenced series between them (Figure S1)."

"The sensitive up-and-down changes in ELISA-SAAs rather than in LTI-SAAs might have resulted in no significant correlation between the differenced series of IDP and ELISA-SAAs (Figure 1S)."

3.3. Literature Integration

 Comparative Context: While the discussion does reference analogous human and equine conditions, it would benefit from more integration with broader marine mammal literature.

Suggestion: Include some reference ranges for the different markers (in all type of animals) and add more comparative data from similar studies in other cetaceans or pinnipeds, if available.

Authors’ response: Thank you for the reviewer’s valuable advice. As the reviewer pointed out, we think that the discussion about the comparative context was not enough. However, if we extend the discussion further to other terrestrial mammals and other types of marine mammals, we are afraid that the focus of the discussion becomes unclear. Therefore, we have thoroughly checked similar studies in cetaceans and found additional three papers (ref. no. 9, 11, and 12 in the revised paper) that reported stranded cases of cetaceans with muscular damage and other complications. Therefore, we added these three references to our paper and discussed more about muscular damage among cetaceans, based on the total five papers (ref. no. 8–12) cited in this paper. We added a new paragraph to the Discussion section to describe these issues.

"Rhabdomyolysis in captive dolphins residing in aquariums and institutes has rarely been reported; therefore, ours may be the first reported captive case that was analyzed in detail clinically, clinicopathologically, and histopathologically [13]; however, we have personally heard of other cases of suspected rhabdomyolysis in captive dolphins. In ad-dition to our case, almost all reported cetacean cases with rhabdomyolysis or capture myopathy were live-stranded cetaceans, including various dolphins and whales that were not cured and eventually died [8–12]. The histopathological findings in all these cases [8–12] and the clinicopathological findings obtained from the blood samples of three cases [10–12] indicated that all the stranded cetaceans had severe skeletal muscle damage and complications, including myoglobinuric nephrosis, AKI, and cardiac dysfunction. Based on these findings, it is difficult to cure these animals. However, early diagnosis and treatments using muscle protectants and antioxidative agents may cure mild or moderate cases and contribute to increasing the survival rates of live-stranded cetaceans, possibly improving the welfare of captive dolphins and wild cetaceans. Further studies are re-quired to evaluate the efficacy of these treatments in live-stranded cases."

Another reviewer (Reviewer 2) requested us to add the description of dolphin welfare in the Introduction section. Therefore, in response to this additional ultimate goal (welfare) in the Introduction section, we include the welfare to the new paragraph about other live-stranded cetaceans in the Discussion section. Please see the Introduction section as well.

Ethical and Transparency Standards

 Ethical approval is appropriately noted.

Funding and potential conflicts of interest are transparently disclosed.

Authors’ response: Thank you for the reviewer's comments about these issues.

Conclusion and Recommendation

 The manuscript presents important preliminary findings on the potential of IDPs and SAA as early biomarkers for skeletal muscle damage in dolphins. While limited by its single-subject design, the study is a valuable contribution to marine veterinary diagnostics.

Authors’ response: We appreciate the reviewer's concern. As we mentioned above, we additionally described the limitation and exploratory nature of the findings and follow-up studies in the conclusion section as follows.

"However, the data from a single case in this study prevented this observation from being fully generalizable; therefore, further studies are required to confirm the clinical utility of IDPs and SAA by increasing the sample size of cases with muscular diseases."

P.S.

 Please make sure that the single error that I found in the article (pdf attached) is addressed

Authors’ response: Thank you for finding our error. We apologize for our misunderstanding. We changed "cicatrices" to "scars".

Round 2

Reviewer 1 Report

Comments and Suggestions for Authors

Thanks for the corrections and effort by the authors' team. No excuse for their response and correction at this version. However, I still think studying a single animal replicate must be deemed a case report. I am leaving it to the editor for further decision, as journals must have their guideline for defining a full-length manuscript and case report, particularly while dealing with giant mammals like dolphins and whales.   I feel comfortable keeping myself out of this debate.

Thank you.